# Gaussian Curvature Entropy for Curved Surface Shape Generation

**DOI:** 10.3390/e22030353

**Published:** 2020-03-18

**Authors:** Akihiro Okano, Taishi Matsumoto, Takeo Kato

**Affiliations:** 1School of Integrated Design Engineering, Graduate School of Keio University, Yokohama 223-8522, Japan; aokano13@keio.jp (A.O.); bananboba@keio.jp (T.M.); 2Department of Mechanical Engineering, Keio University, Yokohama 223-8522, Japan

**Keywords:** complexity, entropy, gaussian curvature, generative design

## Abstract

The overall shape features that emerge from combinations of shape elements, such as “complexity” and “order”, are important in designing shapes of industrial products. However, controlling the features of shapes is difficult and depends on the experience and intuition of designers. Among these features, “complexity” is said to have an influence on the “beauty” and “preference” of shapes. This research proposed a Gaussian curvature entropy as a “complexity” index of a curved surface shape. The proposed index is calculated based on Gaussian curvature, which is obtained by the sampling and quantization of a curved surface shape and validated by the sensory evaluation experiment while using two types of sample shapes. The result indicates the correspondence of the index to perceived “complexity” (the determination coefficient is greater than 0.8). Additionally, this research constructed a shape generation method that was based on the index as a car design supporting apparatus, in which the designers can refer many shapes generated by controlling “complexity”. The applicability of the proposed method was confirmed by the experiment while using the generated shapes.

## 1. Introduction

In designing shapes of products, the overall shape features that emerge due to combinations of shape elements, such as points and lines, are important [1]. These features include “order” and “complexity”, and their control/adjustment is difficult and it depends on the experience and intuition of designers. There are some studies on the relationship between these features and the “beauty” or “preference” of shapes. Birkhoff carried out sensory evaluation while using polygon shapes and argued that the “beauty” of shapes is the ratio of “order” to “complexity” [2]. Eysenck conducted regression analysis for the beauty of polygon shapes and confirmed positive correlation between “complexity”, “order”, and “beauty” [3]. Berlyne insisted that the relationship between “complexity” and “preference” is expressed as an inverted U-shape [4]. Some researchers have confirmed this relationship. Munsinger et al. carried out sensory evaluation about “preference” of 2D shapes and indicated that the shapes having moderate number of vertices are the most preferable [5]. Hung et al. analyzed relationship between “complexity” and “aesthetic preference” and insisted that the relationship is expressed as an inverted U-shape [6].

From the above, since “complexity” is said to have an influence on “beauty” and “preference” of shapes, there are many studies regarding the quantification of “complexity” of curves and curved surface shapes, which are important in designing shapes. For example, Vitz defined the number of lines as “complexity” and analyzed the relationship between “preference” and the number of lines in black line drawings that are generated by the random-walk method [7]. Backes et al. proposed a “complexity” index by employing pixel intensity values of grayscale images and fractal dimension, and proposed a method for image analysis and retrieval [8]. Wang et al. defined the “complexity” of 3D shapes as the difference between outlines of shapes viewed from various angles and proposed an index based on this definition [9,10]. In addition, since Farin argued that curvatures on shapes enables detection of slight changes in shape [11], researches about “beauty” of curves and automatic generation of beautiful curves employed curvature [12,13,14,15]. Similarly, some of the researches applied curvature to quantify “complexity” of curves and curved surface shapes. Ujiie et al. proposed the total absolute curvature and curvature entropy as “complexity” indices of curves and constructed a curve generation method that was based on the indices [1,16]. In addition, Matsumoto et al. proposed total absolute Gaussian curvature as a “complexity” index of curved surface shapes and confirmed the correspondence to the sensory evaluation values about “complexity” [17]. However, the index cannot evaluate the “complexity” of shapes with the same number of concavities, since the index evaluates the number of concavities as “complexity”. Hence, the index sometimes fails to evaluate “complexity” of shapes having small concavities that are difficult to recognize, such as automobiles. Expanding curvature entropy to evaluate curved surface shapes might solve this.

This research aims to propose a “complexity” index of curved surface shapes while using Gaussian curvature entropy and construct a curved surface shape generation method that is based on the index to support the product design. This paper is organized, as follows. Chapter 2 presents the proposal of Gaussian curvature entropy as a “complexity” index of curved surface shapes. Chapter 3 describes the construction of a curved surface shape generation method that is based on the index and its validation experiment. Chapter 4 summarizes the achievements and future tasks.

## 2. Gaussian Curvature Entropy 

A conventional “complexity” index of curved surface shapes fails to evaluate shapes with small concavities, as is mentioned in the previous section. This research aims to solve the problem by adopting the information entropy, which evaluates “complexity” to evaluate the “complexity” of curved surface shapes.

### 2.1. Construction of Gaussian Curvature Entropy

This section describes the definition and calculation method of Gaussian curvature entropy. Gaussian curvature entropy is the information entropy in one-dimensional Markov chain of Gaussian curvature. Gaussian curvature *K* expresses the curve at arbitrary point *p* on a curved surface, and it is calculated as:*K* = *κ*_max_*κ*_min_(1)
where *κ*_max_ and *κ*_min_ are principal curvatures (i.e., the maximum and minimum normal curvature) at *p* (Figure 1). Information entropy *H* expresses the randomness of information [18], and it is calculated as:(2)$$H=-\overset{n}{\underset{i=1}{\sum }}p_{i}\log_{2}p_{i}$$
where *n* is the number of states and *p*_i_ is the occurrence probability of state *s_i_*. *H* becomes higher when the occurrence probability of each state is similar. Note that the occurrence probabilities usually depend on the surrounding states, since there tends to be correlation between each state. This stochastic process considering surrounding states is called the Markov chain. Especially when occurrence probabilities depend on neighboring states, the process is called first order Markov chain. The information entropy *H*’ of the process is calculated as:(3)$$H’=-\overset{n}{\underset{i=1}{\sum }}\overset{n}{\underset{j=1}{\sum }}p_{i}p_{i,j}\log_{2}p_{i,j}$$
where *p_i_*_,*j*_ is the transition probability of state *s_i_* to state *s_j_*. *H*’ becomes higher when the occurrence probability of each state and transition probability between each state are similar. This research employed *H*’ of Gaussian curvature, just as the previous research [1]. Note that, in this research, the transition of Gaussian curvature is expressed as a transition of states in order to calculate *H*’ of Gaussian curvature. Calculation method of the index is explained, as follows. Note that, sampling and quantization are required to allot Gaussian curvature to the states.

(i) Sample a curved surface shape by dividing the entire shape with equilateral triangles with the same area using the Advancing Front Method (Figure 2a) [19]. Note that the triangles generated by the method tend to have similar areas and be close to equilateral triangles. The vertices of the triangles are called sampling points and the Gaussian curvature *K* at point *v* is approximately calculated, as [20]:(4)$$K≅\frac{2\pi -\sum_{t=1}^{m}\alpha_{t}}{\frac{1}{3}\sum_{t=1}^{m}A_{t}}$$
where, *f_t_* (*t* = 1, 2, …, *m*) is a triangle neighboring *v*, *A_t_* is area of *f_t_*, and α*_t_* is angle of *f_t_* at *v* (Figure 3). To get rid of difference in *K* because of difference in *A_t_* between shapes, the dimensionless Gaussian curvature *K*’ is calculated by using the maximum diameter *D* of the curved surface shape, as (Figure 2a):*K’* = *KD*^2^(5)

(ii) Set the parameters for deviation *E* and the state number *V* to define the states of *K*’. Subsequently, quantize *K*’ (i.e., allot *K*’ to each state, as in Figure 2b). *K*’ is quantized based on Equation (6), as:(6)$$s_{1}\;(K’\;<-E),\;s_{2}\;(-E\;≦\;K’\;<-E+\;\Delta E),\;s_{3}\;(-E+\;\Delta E\;≦\;K’\;<-E+2\Delta E),\;\ldots ,\;s_{V-1}\;(E\;-\;\Delta E\;≦\;K’\;<E),\;s_{V}(K’≧E)\;\Delta E=\frac{2E}{V-2}$$

For example, the results of the quantization when {*E*, *V*} = {30, 3} and {*E*, *V*} = {15, 15} are shown in Figure 4a,b, respectively. In Figure 4a, the points at a concavity or convexity are allotted to the second state *s*_2_ as same as points on a plane. Whereas, Figure 4b shows that those points are allotted to different states based on their values of *K*’. Note that, methods to set parameters, which enables calculating entropy without bias, is proposed [21]. However, it is necessary to carry out a sensory evaluation to set parameters since it is important to consider human perceived “complexity” in this research. The setting of parameters without carrying out a sensory evaluation is a future task.

(iii) Calculate the occurrence probability and transition probability of each state (Figure 2c). Occurrence probability *p_i_* is calculated, as:(7)$$p_{i}=\frac{N_{i}}{N}$$
where, *N* is the number of sampling points on the curved surface shapes and *N_i_* is the number of them allotted to the *i*-th (*i* ∈ {1, 2, …, *V*}) state *s_i_*. Afterwards, transition probability *p_i_*_,*j*_ is calculated as:(8)$$p_{i,j}=\frac{N_{i,j}}{N_{i,\mathrm{neighbor}}}$$
where, *N_i_*_,neighbor_ is the number of points neighboring on a point allotted to state *s_i_* and *N_i_*_,*j*_ is the number of transitions from *s_i_* to *j*-th (*j* ∈{1, 2, …, *V*}) state *s_j_*. 

(iv) Gaussian curvature entropy *H*_G_ is calculated while using *p_i_* and *p_i,j_*. At first, information entropy in one-dimensional Markov chain is calculated using Equation (2). Afterwards, *H*_G_ is obtained by dividing the information entropy by the maximum entropy, as:(9)$$H_{G}=-\frac{1}{\log_{2}V}\overset{V}{\underset{i=1}{\sum }}\overset{V}{\underset{j=1}{\sum }}p_{i}p_{i,j}\log_{2}p_{i,j}$$

*H*_G_ ranges between 0 and 1 and becomes 0 when every sampling point on a curved surface shape is allotted to the same state and increases according to the variety of the states that the points are allotted to.

Figure 5 shows comparison of Gaussian curvature entropy *H*_G_ and total absolute Gaussian curvature *I* that was proposed by Matsumoto et al. Note that, calculation method of *I* is described in Appendix A. The values of *I* are 1 on both shapes since the shapes have no concavities. On the other hand, the value of *H*_G_ is 0 on shape A, because the value of *K* is constant on the shape, while the value of *H*_G_ is 0.23 on shape B when {*E*, *V*} = {20, 15} since the value of *K* is various on the shape. Consequently, it seems that Gaussian curvature entropy can evaluate “complexity” due to changes in Gaussian curvature, which is not expressed as the number of concavities.

### 2.2. Experiment for Validation of Gaussian Curvature Entropy

This section illustrates the experiment to examine the correspondence of Gaussian curvature entropy to human perceived “complexity”.

#### 2.2.1. Experimental Methods

1. Sample Shapes

The experiment utilized to types of samples shapes. The shapes are obtained by extrusion and rotation, which are common operations for generating shapes in 3DCAD.

a. Sample Shapes A (Shapes obtained by extrusion)

Sample shapes A are created from 2D shapes while using extrusion. In the experiment, 15 shapes (Figure 6a) are randomly extracted from 25 shapes used in the conventional research [1] and extruded in a direction perpendicular to the shapes. Note that the extruded distance is set to twice as long as the maximum diameter of each shape (Figure 6b).

b. Sample Shapes B (Shapes Obtained by Rotation)

Sample shapes B are created using rotation. 15 shapes are randomly extracted in the same manner, and the right half of the shapes (Figure 7a) are rotated 360 degrees around the axis passing through the center of gravity on each shape (Figure 7b).

2. Experimental Conditions

Evaluation method: This experiment adopts a five-point Likert scale (1: “not complex”, 2: “slightly complex”, 3: “fairly complex”, 4: “complex”, and 5: “very complex”) to obtain sensory evaluation values about complexity in samples shapes.Sample shapes displaying method: White sample shapes on a black background are simultaneously displayed on a seven-inch tablet device. During the experiment, these shapes are rotated at a constant speed (*x* axis: 0 rpm, *y* axis: 0 rpm, *z* axis: 12 rpm) with the center of gravity as the axis, so that the appearance of all concavities can be observed.Presentation method: The distance between the eyeball of a participant and the device was set to 500 mm. 15 sample shapes on the display simultaneously enter the field of view.Participant: 30 participants (23 men and seven women), ranging in age from 18 to 54 years (M = 25.5, SD = 7.78).

#### 2.2.2. Experimental Results and Discussion

1. Results

a. Results of Sample Shapes A

Figure 8 shows the relationship between parameters (*E* and *V*) and the coefficient of determination *R*^2^ of logarithmic approximation between the sensory evaluation values about “complexity” and the value of *H*_G_ of sample shapes A. Note that logarithmic approximation is applied based on Fechner’s law, which indicates the relationship between human sensitivity and stimuli using logarithmic function. *R*^2^ is higher when *E* is between [20, 300] and *V* is an odd number, as shown in Figure 8. Moreover, the value of *E* with high *R*^2^ becomes larger as *V* gets larger. Figure 9 shows the relationship between the sensory evaluation values and *H*_G_ when *R*^2^ is the largest value ({*E*, *V*} = {200, 11}). This figure shows the *R*^2^ is 0.85 and the correspondence between *H*_G_ and the sensory evaluation values. However, there are shapes whose values of *H*_G_ differs while their sensory evaluation values are close, such as shapes 8 and 14 (Figure 9).

b. Result of Sample Shapes B

Just the same as sample shapes A, Figure 10 shows the relationship between parameters (*E* and *V*) and *R*^2^ of sample shapes B. *R*^2^ is higher when *E* is between [20, 70] and *V* is an odd number, as shown in Figure 10. As same as sample shapes A, the value of *E* with high *R*^2^ becomes larger as *V* gets larger. Figure 11 show the relationship between the sensory evaluation values and *H*_G_ when *R*^2^ is the largest value ({*E*, *V*} = {50, 19}). The *R*^2^ is 0.83 and the correspondence between *H*_G_ and the sensory evaluation values is confirmed, as shown in Figure 11. However, there are shapes whose relationship between the sensory evaluation values and *H*_G_ is reversed, such as shapes 9 and 11 (Figure 11).

2. Discussion

a. Discussion about Sample Shapes A

(i) Shapes whose values of *H*_G_ differs, while their sensory evaluation values are close

Figure 12 shows shapes 8 and 14. Between these shapes, values of *H*_G_ are higher in shape 8, while sensory evaluation values are close. Shape 8 has a concavity where values of *K*’ are low and this makes the value of *H*_G_ lower than shape 14, which has no concavities, as shown in Figure 12. However, the concavity of shape 8 is bent at a right angle, which is often seen in general industrial products. As a result, it seems that the concavity does not affect human perceived “complexity” and the sensory evaluation value of shape 8 become low and closer to that of shape 14. This might be improved by considering right angles in the calculation of *H*_G_.

(ii) Setting of Parameters

Figure 13 shows an example of quantization when {*E*, *V*} = {200, 11}, which presents the highest value of *R*^2^. First, the setting of *E* is discussed, as follows. In shape 11, points at sharp convexities are allotted to *s*_11_ while points at concavities are allotted to *s*_1_. In shape 13, points in gradual convexities are allotted to *s*_9_ or *s*_8_, while points in gradual concavities are allotted to s_4_ or *s*_3_. In shape 6, points located out of convexities are allotted to *s*_6_. Therefore, it seems that various convexities and concavities of sample shapes A are allotted to different states that are based on their sharpness by setting parameters at {*E*, *V*} = {200, 11}.

Next, setting of *V* is discussed, as follows. Figure 14 shows the difference in quantization of *K*’ when *V* = 3 or 4 in shape 6 of sample shapes A. When *V* = 3, the points on planes (surface where value of *K*’ is nearly 0) are allotted to the same state, while points on corners are allotted to other states and the value of *H*_G_ is 0.071. When *V* = 4, the points on planes are allotted to two states because of minute unevenness and the value of *H*_G_ is 0.520. Therefore, it is appropriate to set *V* at an odd number to prevent overestimating minute unevenness.

Afterwards, the tendency that the value of *E* with high *R*^2^ becomes larger as *V* gets larger is discussed. For example, the value of *R*^2^ becomes high not only when {*E*, *V*} = {200, 11} but also when {*E*, *V*} = {300, 15} or {300, 17}, as shown in Figure 8. This is because boundaries between states are similar among these settings. For example, there are boundaries at *K*’ = 22, 67, 111, 166, and 200 when {*E*, *V*} = {200, 11}. On the other hand, boundaries are at *K*’ = 23, 69, 115, 162, and 208 when {*E*, *V*} = {300, 15} and *K*’ = 20, 60, 100, 140, and 180 when {*E*, *V*} = {300, 17}. Therefore, the tendency occurred because boundaries of *K*’ are similar among those settings of parameters. Further, cross-validation is carried out in Appendix B to confirm the tendency.

b. Discussion about Sample Shapes B

(i) Shapes whose relationship between the sensory evaluation values and *H*_G_ is reversed

Figure 15 shows shapes 9 and 11. Between these shapes, the values of *H*_G_ were higher in shape 9 while the sensory evaluation values were higher in shape 11. Shape 9 has a sharp edge and points neighboring the edge are allotted to various states, as shown in Figure 15. As a result, the value of *H*_G_ rises. It seems that this is because the sampling points are not distributed along the edge and the values of *K*’ at the points are varied. Therefore, this can be solved by distributing sampling points along the edge during sampling. 

(ii) Setting Parameters

Figure 16 shows the examples of quantization when {*E*, *V*} = {50, 19}, which presents the highest value of *R*^2^. In shape 8, points at sharp convexities are allotted between *s*_19_ and *s*_17_, while points at sharp concavities are allotted to *s*_1_. In shape 5, the points at gradual convexities are allotted between *s*_15_ and *s*_12_, while points at gradual concavities are allotted to *s*_9_. In shape 6, the points at plane surfaces are allotted to *s*_10_. This suggests that the convexities and concavities of sample shapes B are allotted to different states based on their sharpness by setting parameters at {*E*, *V*} = {50, 19}. Note that the reason why *V* becomes an odd number is to prevent overestimating minute unevenness, the same as sample shapes A.

At last, the difference between *H*_G_ and total absolute Gaussian curvature *I* is discussed. Figure 17 shows shapes 10 and 13 of sample shapes B as shapes corresponding to *H*_G_ rather than *I*. In these shapes, sensory evaluation values are 3.38 and 3 and the values of *H*_G_ are 0.210 and 0.152, respectively. However, values of *I* are 1.46 and 1.54, which are not corresponding to the sensory evaluation values. It seems that this is because *I* evaluates the number of concavities as “complexity” and *I* cannot evaluate the size of concavities. On the other hand, values of *H*_G_ differ because the values of *K*’ differ in concavities with different size. Therefore, it seems that *H*_G_ overcomes a shortcoming of *I*, which is unable evaluate “complexity” between shapes having the same number of concavities. Note that the tendency that the value of *E* with high *R*^2^ becomes larger as *V* gets larger is because the boundaries of *K*’ are similar among those settings of parameters, as same as sample shapes A. Further, cross-validation is carried out in Appendix B to confirm the tendency.

Figure 18 shows shapes 4 and 10 of sample shapes A as shapes corresponding to *I* rather than *H*_G_. In these shapes, sensory evaluation values are 2.71 and 2.79 and values of *I* are 1.59 and 1.66, respectively. However, values of *H*_G_ are 0.044 and 0.057, which do not correspond to sensory evaluation values. It seems that this is because states of sampling points on edges are allotted to different states. On an edge of shape 4 (Edge A in Figure 18a), the edge is straight,, except for a bent and the state of the points are *s*_6_, since the value of *K*’ is close to 0. On the other hand, on an edge of shape 10 (Edge B in Figure 18b), the edge is slightly bent and the state around the center of is *s*_5_, while the state close to the corner is between *s*_7_ and *s*_11_. Consequently, the variance of states is larger and the value of *H*_G_ is higher in shape 10. Therefore, it seems that *H*_G_ sometimes overestimates a slight difference in a value of *K*’.

## 3. Shape Generation Method

### 3.1. Construction of Shape Generation Method

This section describes the construction of curved surface generation method that is based on Gaussian curvature entropy.

At first, an expression method of a curved surface shape and an optimization method of a shape based on Gaussian curvature entropy is explained. This research employed Non-Uniform Rational B-Spline (NURBS) surface as the expression method. NURBS surface is generated based on position ***Q****_ab_* (*a* = 1, 2, …, *k*, *b* = 1, 2, …, *l*) weight *w_ab_* of controlling points distributed on the *k* × *l* grid [22]. Note that the NURBS surface has higher flexibility and it does not require constraining positions of controlling points in order to connect multiple surfaces with their tangents continuous. Therefore, it might be possible to generate curved surface shapes effectively by applying NURBS surface.

In addition, this research employed Particle Swarm Optimization (PSO) as a method to optimize ***Q****_ab_* and *w_ab_* of NURBS surface based on the value of *H*_G_. Note that PSO is capable of solving problems, such as minimization of a multimodal function and optimization of a combination [23].

Figure 19 shows the summary of the proposed method. The procedure of the method is explained, as follows.

(i)Set the initial shape (Figure 19a). Subsequently, the initial shape is expressed while using the NURBS surface and ***Q****_ab_* and *w_ab_* of the surface is defined as the position vectors of particles in PSO (Figure 19b). Note that ***Q****_ab_* is expressed by polar coordinates whose origin is the position of the controlling point at the initial shape.(ii)Set *H*_G,target_ (targeted value of *H*_G_) and *f*_u_ (allowable difference between *H*_G_ and *H*_G,target_) as a condition of a candidate for solution.(iii)Generate shapes that are based on a position vector of particles renewed by movement of particles. Subsequently, *H*_G_ and fitness *f* of the shapes is calculated. Note that the movable range of *r_ab_* during movement is the half of the distance to the closest controlling point. In addition, this research set the range of *w_ab_* to 0.5 ≤ *w_ab_* ≤ 2.0, since the shape transforms drastically by changing the value in the range. Finally, if *f* of a generated shape is lower than *f*_u_, the shape is output as a candidate for solution. On the other hand, the movement of particles generates other shapes if there are no shapes meeting the condition.

Note that the proposed method cannot generate a shape that is based on a grid whose controlling point positions are inverted to prevent the generation of shapes with crossed surface. This means that there is a possibility that the generated shapes lack diversity.

### 3.2. Experiment for Validation of Shape Generation Method

This section explains the experiment to examine human perceived “complexity” of shapes that are generated by the proposed method.

#### 3.2.1. Experimental Methods

1. Generated Shapes

In this research, the method was applied to generate the shapes of Coupe-typed automobiles. At first, the NURBS surface was generated as the initial shape based on controlling points (Figure 20). Note that the movement of the points was constrained, as shown in Figure 20. The value of *H*_G_ at the shape was 0.322 and shapes are generated by setting five levels of *H*_G,target_ at 0.30, 0.38. 0.46, 0.54, and 0.62. Note that *f*_u_ was set at 0.004 and 15 shapes (three shapes at each level) were generated. The reason of setting *f*_u_ at 0.004 was because of the difference between the levels of *H*_G,target_ was 0.08. Under this setting, the *H*_G_ of the generated shape should be between [*H*_G,target_ – 0.04, *H*_G,target_ + 0.04]. Subsequently, *f*_u_ was set at 0.004, which was 1/10 of 0.04. In addition, the parameters of *H*_G_ were set at {*E*, *V*} = {2, 3}. This is because, in the preliminary experiment, the relationship between the sensory evaluation values about “complexity” and the value of *H*_G_ of shapes generated by random transformation of the initial shape is examined and the *R*^2^ of logarithmic approximation on the relationship is maximized when {*E*, *V*} = {2, 3}. Figure 21 shows the generated shape examples.

2. Experimental Conditions

Evaluation method: Just the same as Section 2.2.1.Sample shapes displaying method: White generated shapes on a black background were simultaneously displayed on a 13.3-inch laptop computer (Figure 22). During the experiment, each shape is rotated at a constant speed (*x* axis: 0 rpm, *y* axis: 0 rpm, *z* axis: 10 rpm), so that the appearance of all concavities can be observed.Presentation method: The distance between the eyeball of a participant and the computer was set to 500 mm. 15 sample shapes on the display simultaneously enter the field of view.Participants: 40 participants (36 men and four women) that ranged in age from 16 to 61 years (M = 23.9, SD = 7.67).

#### 3.2.2. Experimental Results and Discussion

1. Results

Figure 23 shows the relationship between the sensory evaluation values about “complexity” and the value of *H*_G_. The *R*^2^ of logarithmic approximation on the relationship is 0.71, and the correspondence between *H*_G_ and the sensory evaluation values is confirmed. However, there are shapes that deviate from the approximate curve, such as shape 10, 11, and 15.

2. Discussion

Figure 24 shows the quantization of *K*’ on shapes that deviate from the approximate curve. These shapes have concavities at rear section and sampling points at the concavities are allotted to *s*_1_, as shown in this figure. It seems that the sensory evaluation value of the shape become lower, since it is difficult to recognize concavities at rear section. This is because it is natural to focus on the front section of the shape of an automobile rather than rear section. To examine this, *H*_G_ considering only the front section is calculated and the correspondence to the sensory evaluation value is verified (Figure 25). Consequently, the deviation of shape 10, 11, and 15 from the approximate curve is mitigated and the *R*^2^ of logarithmic approximation is 0.89. Therefore, in calculating *H*_G_, it seems that it is necessary to consider the position of states and this can be achieved by weighing states based on their position.

## 4. Conclusions

This research proposed Gaussian curvature entropy as a “complexity” index of curved surface shapes by expanding curvature entropy, which is a “complexity” index of curves. Subsequently, correspondence between the index and human perceived “complexity” was examined by an experiment while using samples shapes generated by extrusion and rotation. Consequently, the coefficient of determinations between the value of the index and the sensory evaluation values about “complexity” were 0.85 and 0.83, and we confirmed the correspondence between *H*_G_ and the sensory evaluation values. Moreover, knowledge about the parameters of Gaussian curvature entropy was obtained.

Using the proposed index, this research constructed a curved surface shape generation method. The method was applied to generate the shapes of Coupe-typed automobiles and correspondence between the index and human perceived “complexity” of generated shapes was examined. The result shows the coefficient of determination between the value of the index and the sensory evaluation values regarding “complexity” was 0.71, and the usability of the method was confirmed. The result also shows the difficulty to recognize concavities at rear section of the shapes, and this decreases the sensory evaluation values. This means that the human bias focusing on the front section rather than rear section suggested the need to define the weight based on the section of shapes.

The future study should consider the construction of a method for defining the weight based on their position during the calculation of Gaussian curvature entropy and carry out the validation experiment of the curved surface design by designers.

## Figures and Tables

**Figure 1 entropy-22-00353-f001:**
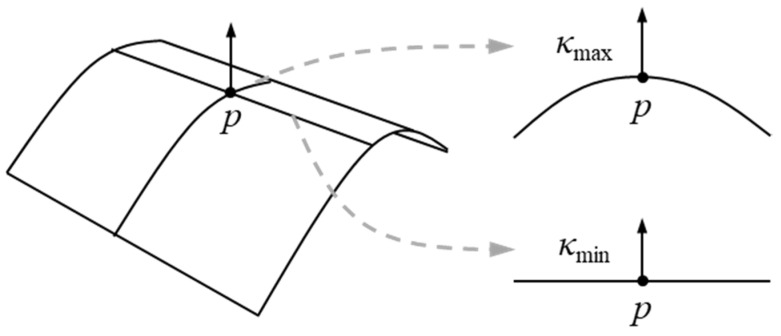
Principal curvatures *κ*_max_ and *κ*_min_ at point *p*.

**Figure 2 entropy-22-00353-f002:**
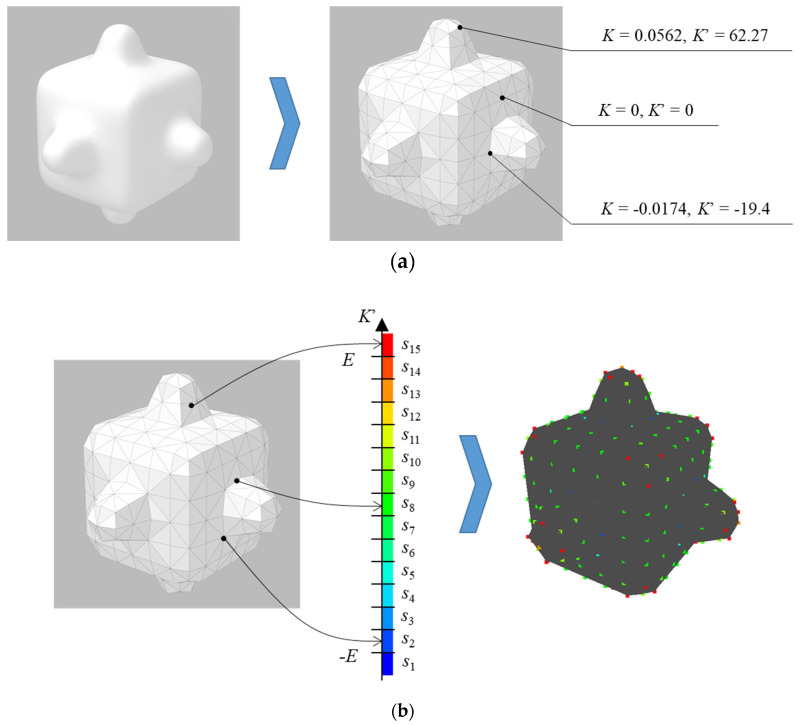
Calculation of Gaussian curvature entropy: (**a**) Division of curved surface and calculation of dimensionless Gaussian curvature; (**b**) Quantization of dimensionless Gaussian curvature; and, (**c**) Calculation of occurrence probability and transition probability.

**Figure 3 entropy-22-00353-f003:**
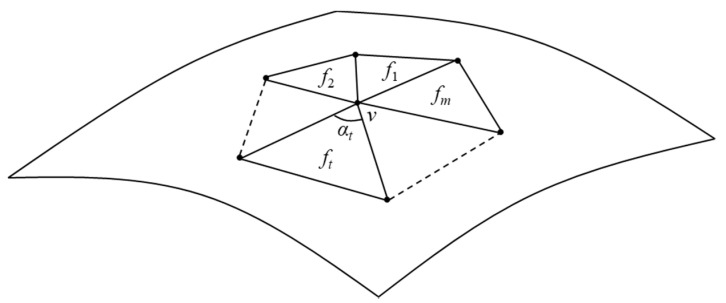
Calculation of Gaussian curvature at sample point *v*.

**Figure 4 entropy-22-00353-f004:**
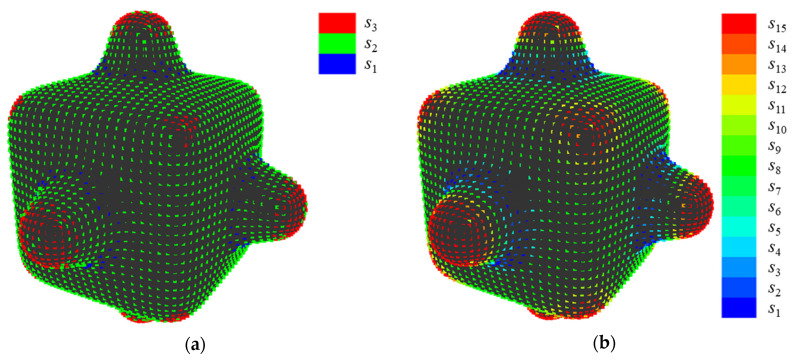
Quantization of dimensionless Gaussian curvature at each point: (**a**) {*E*, *V*} = {30, 3}; and, (**b**) {*E*, *V*} = {15, 15}.

**Figure 5 entropy-22-00353-f005:**
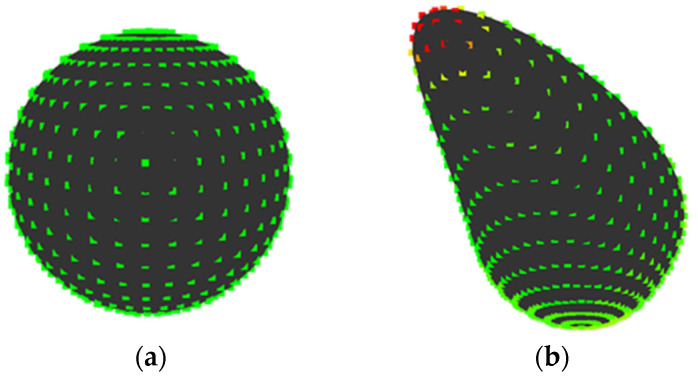
Quantization of dimensionless Gaussian curvature on shapes without concavities when {*E*, *V*} = {20, 15}: (**a**) sphere; (**b**) distorted sphere.

**Figure 6 entropy-22-00353-f006:**
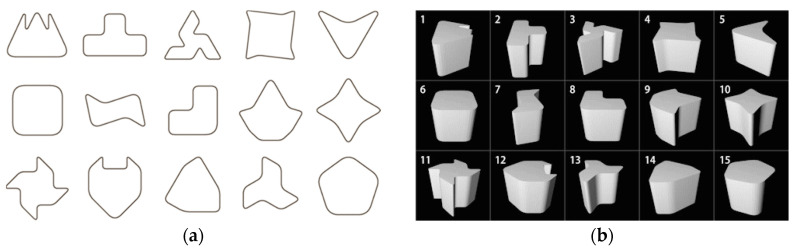
Generation of sample shapes A: (**a**) Curved shapes to extrude; (**b**) Sample shapes A.

**Figure 7 entropy-22-00353-f007:**
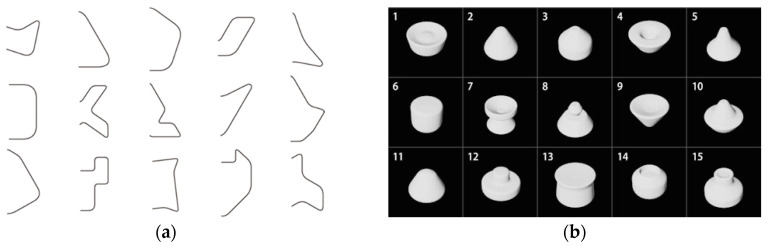
Generation of sample shapes B: (**a**) Curved shapes to rotate; (**b**) Sample shapes B.

**Figure 8 entropy-22-00353-f008:**
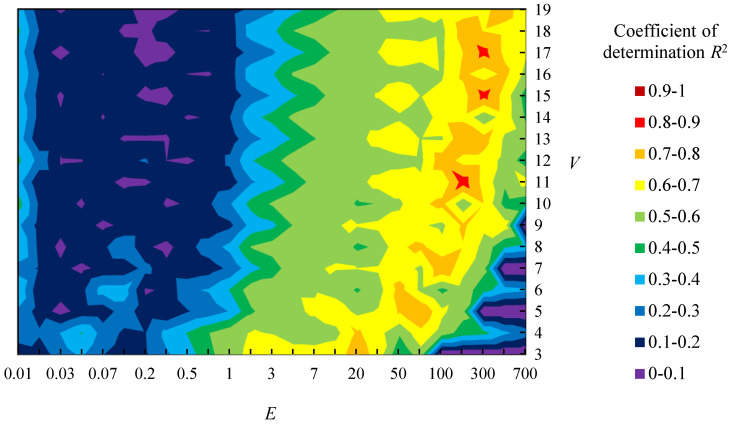
Relationship between parameters (*E* and *V*) and coefficient of determination *R*^2^ in sample shapes A.

**Figure 9 entropy-22-00353-f009:**
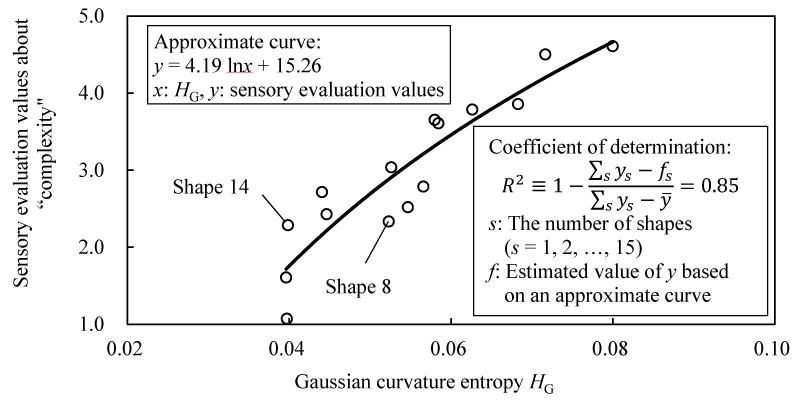
Relationship between Gaussian curvature entropy *H*_G_ and sensory evaluation values about “complexity” in sample shapes A.

**Figure 10 entropy-22-00353-f010:**
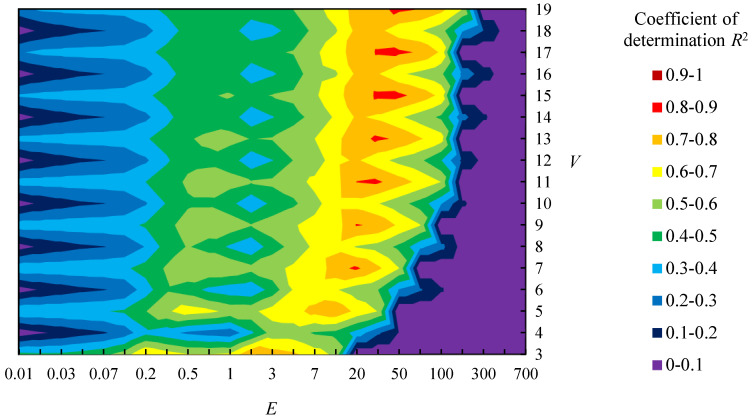
Relationship between parameters (*E* and *V*) and coefficient of determination *R*^2^ in sample shapes B.

**Figure 11 entropy-22-00353-f011:**
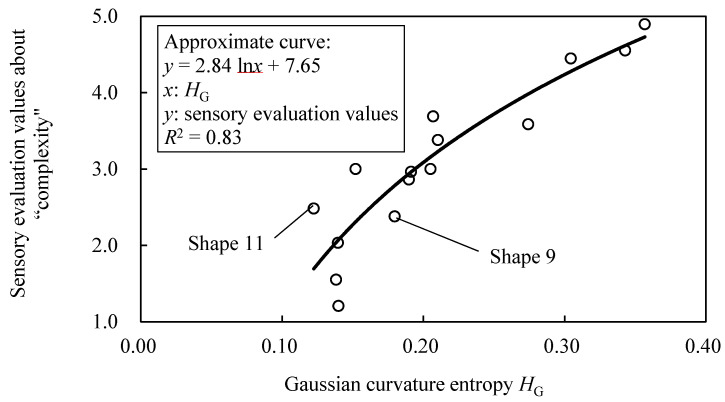
Relationship between Gaussian curvature entropy *H*_G_ and sensory evaluation values about “complexity” in sample shapes B.

**Figure 12 entropy-22-00353-f012:**
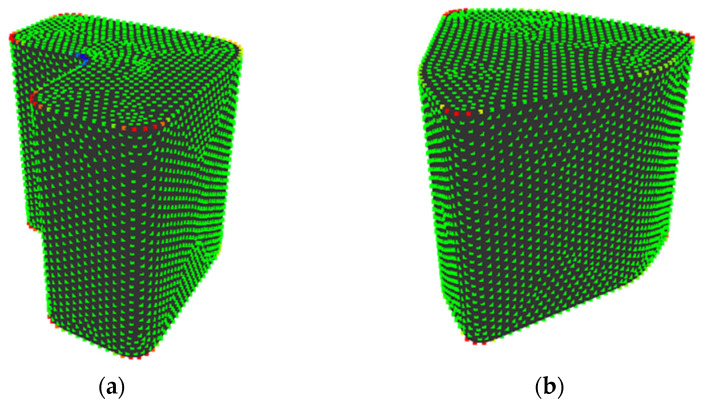
Shapes whose values of *H*_G_ differs while their sensory evaluation values are close: (**a**) shape 8; (**b**) shape 14.

**Figure 13 entropy-22-00353-f013:**
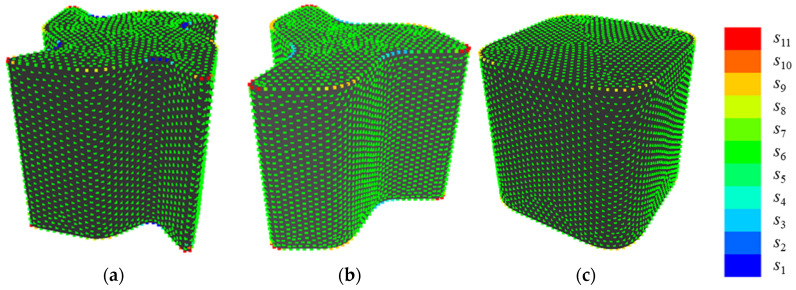
Examples of quantization when {*E*, *V*} = {200, 11}: (**a**) shape 11; (**b**) shape 13; and, (**c**) shape 6.

**Figure 14 entropy-22-00353-f014:**
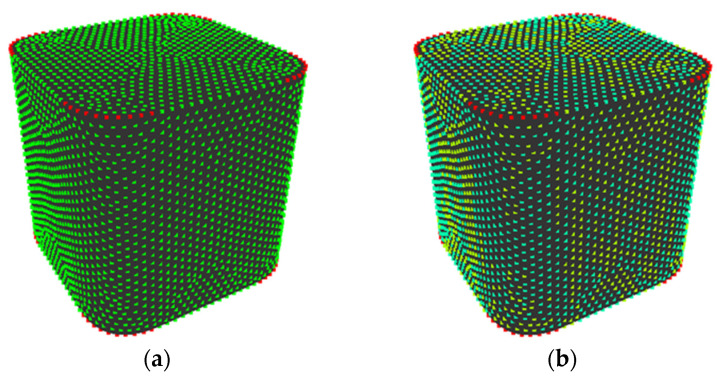
Difference in quantization of dimensionless Gaussian curvature: (**a**) *V* = 3; (**b**) *V* = 4.

**Figure 15 entropy-22-00353-f015:**
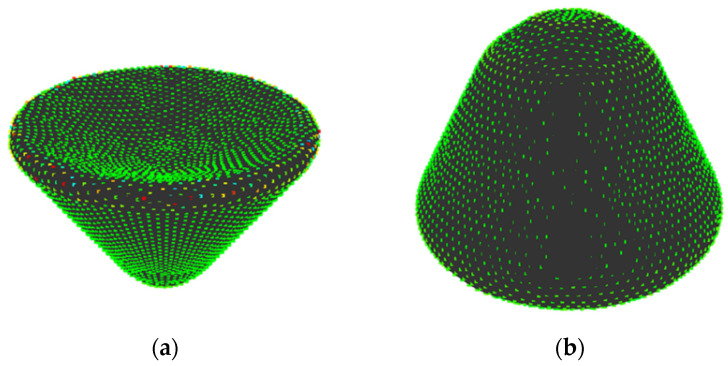
Shapes whose relationship between the sensory evaluation values and *H*_G_ is reversed: (**a**) shape 9; (**b**) shape 11.

**Figure 16 entropy-22-00353-f016:**
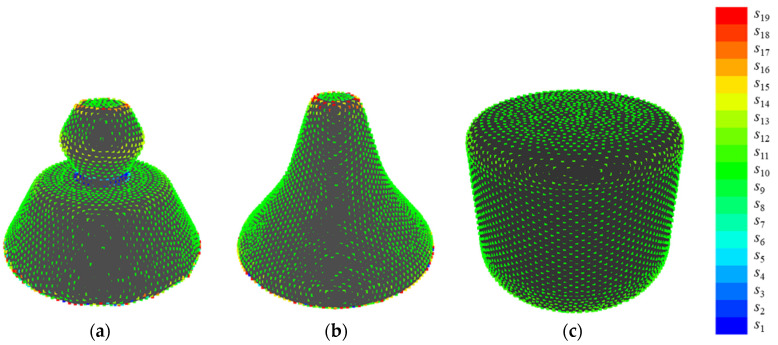
Examples of quantization when {*E*, *V*} = {50, 19}: (**a**) shape 8; (**b**) shape 5; and, (**c**) shape 6.

**Figure 17 entropy-22-00353-f017:**
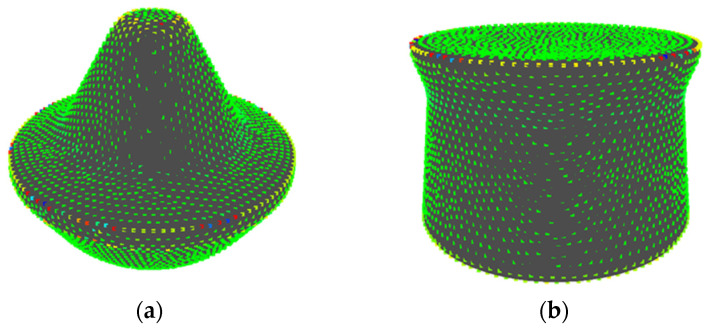
Shapes whose sensory evaluation values correspond to Gaussian curvature entropy rather than total absolute Gaussian curvature: (**a**) shape 10; (**b**) shape 13 in samples shapes B.

**Figure 18 entropy-22-00353-f018:**
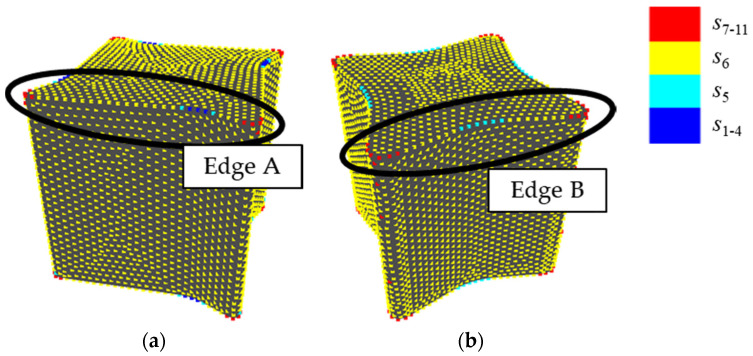
Shapes whose sensory evaluation values correspond to total absolute Gaussian curvature rather than Gaussian curvature entropy: (**a**) shape 4; (**b**) shape 10 in sample shapes A.

**Figure 19 entropy-22-00353-f019:**
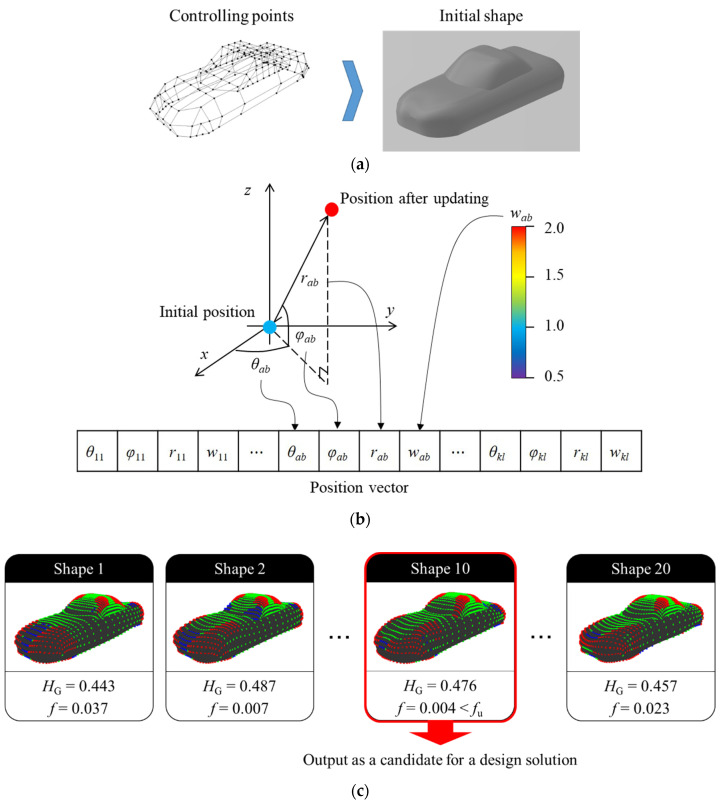
Shape generation method based on Gaussian curvature entropy: (**a**) Setting of an initial shape, *H*_G,target_ and *f*_u_; (**b**) Generation of position vectors; and, (**c**) Generation and output of shapes.

**Figure 20 entropy-22-00353-f020:**
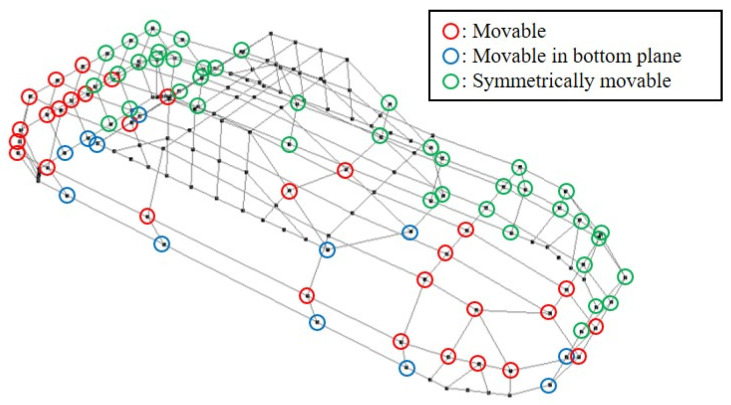
Initial shape of shape generation.

**Figure 21 entropy-22-00353-f021:**
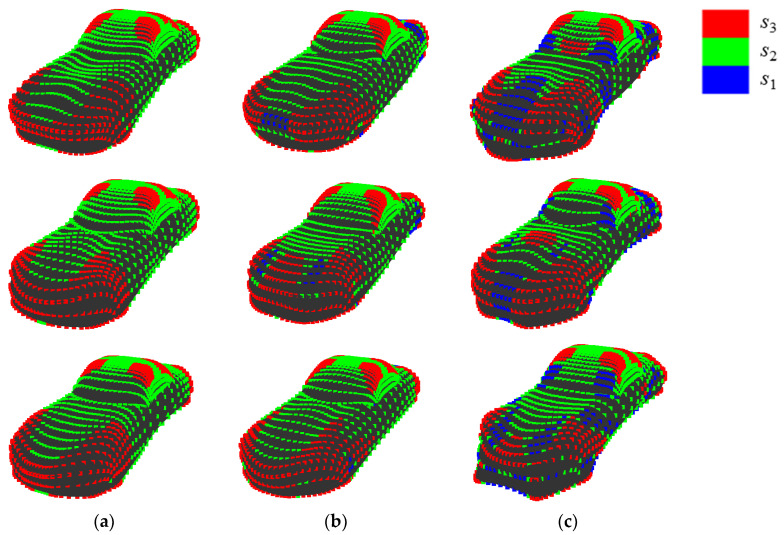
Examples of generated shapes: (**a**) *H*_G_ = 0.30; (**b**) *H*_G_ = 0.46; and, (**c**) *H*_G_ = 0.62.

**Figure 22 entropy-22-00353-f022:**
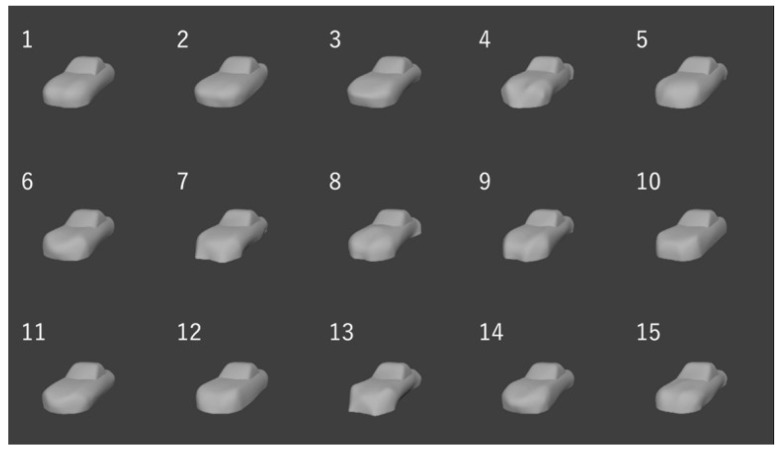
Displaying method of generated shapes.

**Figure 23 entropy-22-00353-f023:**
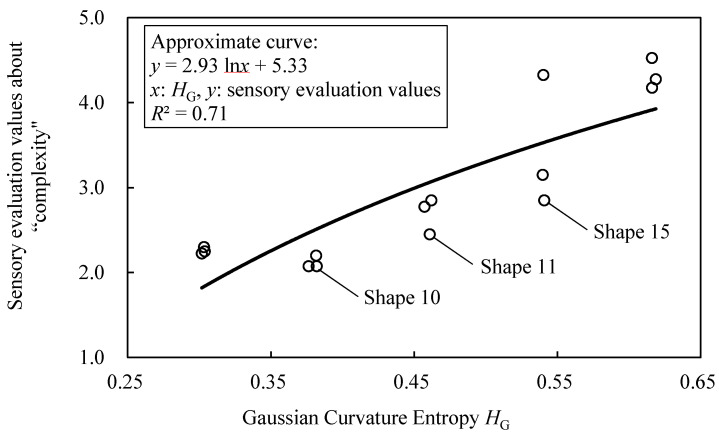
Relationship between Gaussian curvature entropy *H*_G_ and sensory evaluation values about “complexity” in generated shapes.

**Figure 24 entropy-22-00353-f024:**
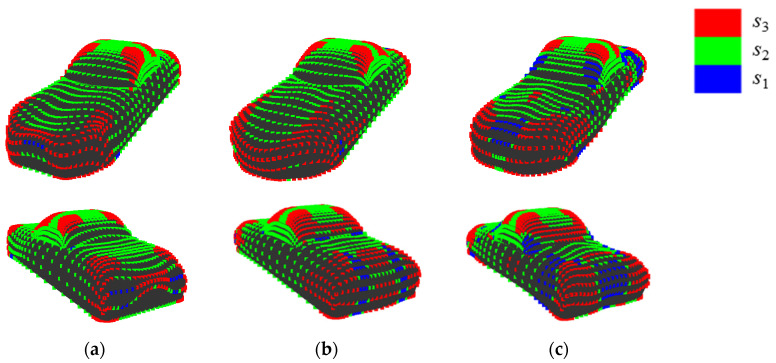
Quantization of dimensionless Gaussian curvature in shapes deviating from approximate curve: (**a**) shape 10; (**b**) shape 11; and, (**c**) shape 15.

**Figure 25 entropy-22-00353-f025:**
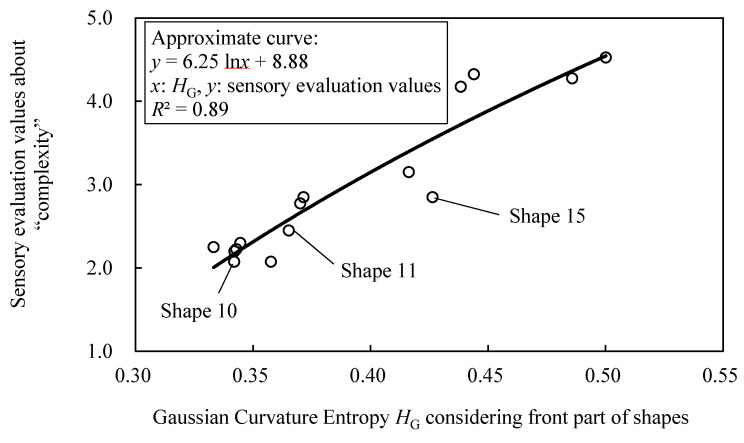
Relationship between Gaussian curvature entropy *H*_G_ considering front part of shapes and sensory evaluation values about “complexity” in generated shapes.

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
