# Peer review of "Gaussian Curvature Entropy for Curved Surface Shape Generation"

_entropy, 2020, doi:10.3390/e22030353_

Round 1

Reviewer 1 Report

This review paper is well written about Gaussian curvature entropy for curved surface shape generation. The proposed method to evaluate a complexity index of curved surface shapes using Gaussian curvature entropy is interesting. The proposed method to generate a curved surface shape based on the index is also interesting. I have some following comments.

It is necessary to explain the meaning of the parameters for deviation E and the state number V.

Why do you use a logarithmic approximation between the sensory evaluation values about “complexity” and the value of HG?

Sec. 3.2.1: Why did you set allowable difference fu at 0.004?

Author Response

Dear reviewer,

Thank you very much for reviewing our paper and giving us insightful comments. We feel the comments are helpful to significantly improve the paper.

Reviewer’s comments

Responses

It is necessary to explain the meaning of the parameters for deviation E and the state number V.

We apologize the lack of explanation about definition of parameters V and E. V is the number of states for quantizing dimensionless Gaussian curvature K’, while E defines quantization of K’ as follows:

s1 (K’ < –E), s2 (–E ≦ K’ < –E + ΔE),

s3 (–E + ΔE ≦ K’ < –E + 2ΔE), …,

sV – 1 (E – ΔE ≦ K’ < E), sV (K’ ≧ E)

E = 2E/(V – 2))

Why do you use a logarithmic approximation between the sensory evaluation values about “complexity” and the value of HG?

This study analyzed the relationship between the sensory evaluation values and HG with logarithmic approximation based on Fechner’s law, which indicates the relationship between human sensitivity and stimuli using logarithmic function as follows;

E = ClogR

where, E is the strength of human sensitivity (sensory evaluation values), C is the constant and R is the strength of stimuli.

The description [p.7, line 156] was added.

Sec. 3.2.1: Why did you set allowable difference fu at 0.004?

In the shape generation, we set the difference between the levels of HG,target at 0.08. Under this setting, HG of generated shape should be between [HG,target – 0.04, HG,target + 0.04]. Then, we set fu at 0.004, which is 1/10 of 0.04.

The description [p.15, line 291] was added.

An attached file is the revised manuscript.

Best regards,   Keio University School of Integrated Design Engineering Akihiro Okano E-Mail: [email protected]

Reviewer 2 Report

In this paper, the authors proposed the Gaussian curvature entropy as a measure for complexity. Then the authors proposed a method for creating surface shapes using the proposed measure. However, the meanings of many parameters in this paper are not clear. For examples;

  1. $E$ is defined as the deviation parameter in page 3. However, they do not provide any mathematical or formal definition of $E$. Therefore, it is not clear exactly what $E$ means.
  2. From page 7, suddenly $R^2$ is used in the figures and related discussions. However, because there is no proper definition, it does not convey any information and thus these discussions cannot be properly judged at all.

Because the above parameters are the main notations and concepts for understanding the whole paper, it is difficult to follow the paper because the relevant definitions are omitted in the paper.

In addition, the difference between the human psychological judgment of complexity and the proposed parameter seems somewhat arbitrary. Especially, the authors need to provide the guarantee of the any (or many) shapes can be treated as in the same way in the section 2.

Other minor comments are as follows;

  1. Is $K$ in the equation (1) the same as $K$ in the equation (4)? If yes, the authors should provide the proof or reference. Otherwise, they should use a different symbol for it.
  2. In the equation (4), are $A_t$ for all $t$ the same? Since at the line 84-85, the triangles have the same area, it seems that the equation (4) can be simplified.
  3. The definition of the total absolute Gaussian curvature $I$ should be defined (with the reference) before used in the paper.

In conclusion, it is necessary to modify the paper as a whole so that it can be understood. Currently, I cannot recommend accepting this paper for publication.

Author Response

Dear reviewer,

Thank you very much for reviewing our paper and giving us insightful comments. We feel the comments are helpful to significantly improve the paper.

Reviewer’s comments

Responses

E is defined as the deviation parameter in page 3. However, they do not provide any mathematical or formal definition of E. Therefore, it is not clear exactly what E means.

We apologize the lack of explanation about definition of parameters V and E. V is the number of states for quantizing dimensionless Gaussian curvature K’, while E defines quantization of K’ as follows:

s1 (K’ < –E), s2 (–E ≦ K’ < –E + ΔE),

s3 (–E + ΔE ≦ K’ < –E + 2ΔE), …,

sV – 1 (E – ΔE ≦ K’ < E), sV (K’ ≧ E)

E = 2E/(V – 2))

From page 7, suddenly R2 is used in the figures and related discussions. However, because there is no proper definition, it does not convey any information and thus these discussions cannot be properly judged at all.

We are sorry for lack of explanation about coefficient of determination R2. R2 indicates how well the approximate curve corresponds to objective variables (sensory evaluation values about “complexity”).

Further, the definition of R2 is added to Figure 9.

In addition, the difference between the human psychological judgment of complexity and the proposed parameter seems somewhat arbitrary. Especially, the authors need to provide the guarantee of the any (or many) shapes can be treated as in the same way in the section 2.

We decided to carry out sensory evaluations with 15 shapes in each sample shape because it was difficult for the participants to evaluate them at the same time if we show more than 15 shapes.

In the condition, we chose two types of sample shapes (made by extrusion and rotation) because these operations are common to generate shapes in 3DCAD (i.e. to be used to design industrial products) as shown in [p.6, line 126].

Would you please tell us if there are any other type of shapes which is suitable/inevitable for the evaluation? Thank you for your important suggestion.

Is K in the equation (1) the same as K in the equation (4)? If yes, the authors should provide the proof or reference. Otherwise, they should use a different symbol for it.

Equation (4) indicates the method to approximately calculate K defined in equation (1).

The reference introducing the approximation is added [p.2, line 86]. In addition, we added approximately equal sign to equation (4).

In the equation (4), are At for all t the same? Since at the line 84-85, the triangles have the same area, it seems that the equation (4) can be simplified.

We apologize the lack of explanation about t. All t is same including At. In addition, we have modified Figure 3 using t.

Further, this study applied Advancing Front Method to divide a curved surface shape. The method divides a shape with equilateral triangles with same area as much as possible. Therefore, there should be triangles which is not equilateral or whose area is not same as others.

The description [p.2, line 85] and the reference is added.

The definition of the total absolute Gaussian curvature I should be defined (with the reference) before used in the paper.

We are sorry for lack of description about total absolute Gaussian curvature I. The description of I was added as Appendix A [p.19, line 396] with the reference [17].

An attached file is the revised manuscript.

Best regards,

Keio University School of Integrated Design Engineering Akihiro Okano E-Mail: [email protected]

Reviewer 3 Report

The authors investigate the relationship between estimated curvature entropy and 'complexity' perception by using rendered 3D shapes. The shapes are tesselated with equilateral triangles. Curvature is estimated from these triangles, then a discrete distribution over triangles is estimated, which is in turn used to estimate curvature entropy. Curvature entropy is used to predict subjective complexity ratings by  selecting the entropy range and discretization steps which yields the best fit using a logarithmic function. Various possible strengths and shortcomings are discussed, then the method is applied to automatic design of a 'complex looking' car. It would have been very interesting to make the connection to 'beauty' or 'preference' that the introduction refers to.

Major comments:

  • Section 2 skips from curvature to entropy without a clear motivation or even a paragraph break. It does become clear later why these two concepts are introduced, but it would improve readability if that was stated at the beginning.
  • line 76: '..time series is called Markov Process." True, but there is no time series here. Instead, the transitions are in an abstract 1-d space. Please rewrite
  • line 84: 'equilateral triangle'. If these triangles are flat, then all angles have to be 60 degrees, which makes formula (4) uninformative. Please clarify
  • line 88: '...because of difference in A_t..' contradicts with line 84:'...triangles with same area...'
  • line 90: define E more clearly (range of H which is discretized). What do you do with values of H outside of [-E,E]?
  • 97: 'where N is the number of sampling points on the curved surface shapes..' please clarify. Do you mean the vertices of the triangles, as defined in line 85?
  • eqn. 7: due to the 2-D surface geometry, there are more transitions than sampling points, if I understand you correctly. Thus, p_i,j will not be a probability. Please clarify, and explain how table 2(c) is computed
  • fig 3: please explain exactly what the angle alpha_1 is here.
  • line 151: why are you using a logarithmic approximation?
  • Section 'Results': you pick the values of  E and V that maximize R^2 of the logarithmic fit.  Please convince the reader that you are not overfitting your data. At the very least, you should present cross-validation results.
  • Section 'Results': given that the results in fig. 9 are cherry-picked post-hoc from the ranges of E,V in your specfic experiment, it is unclear how one could generalize this approach to predict 'complexity' of new shapes. Please demonstrate how you can select E,V, without using perceptual data, e.g. by evaluating the quality of the entropy estimate.
  • Entropy estimates from limited samples are known to be heavily biased, and a range of techniques was developed to deal with that, see e.g. Endres & Földiak 2005 IEEE Trans.Inf.Theo. for an overview. Especially with V=19 (figure 8) such a correction would be in order to reduce noise-induced overfitting.
  • The discussion should be rewritten after the methodology has been debugged and the data re-evaluated.

Author Response

Dear reviewer,

Thank you very much for reviewing our paper and giving us insightful comments. We feel the comments are helpful to significantly improve the paper.

An attached is a response to comments.

Best regards,

Keio University School of Integrated Design Engineering Akihiro Okano E-Mail: [email protected]

Round 2

Reviewer 2 Report

All of my concerns have been resolved. I would like to accept this paper.

Author Response

Dear reviewer,

Thank you very much for reviewing our paper.

Best regards,

Keio University

School of Integrated Design Engineering

Akihiro Okano

E-Mail: [email protected]

Reviewer 3 Report

Thank you for the revision, the manuscript is much improved now.

There are two changes I would strongly recommend before publication

  • remove all references to 'time' when describing the construction of the conditional probability (Markov chain), since time has nothing to do with the described construction.
  • Appendix B is incomplete and the cross-validation procedure is described in a way that I did not understand. In particular, how are the subsamples selected? I can guess what the authors did, but it should be clear to the reader by improving the description with the help of an English language expert.
  • Language should be checked by an expert, especially in the newly added parts

Author Response

Dear reviewer,

Thank you very much for reviewing our paper and giving us insightful comments. We feel the comments are helpful to outstandingly improve the paper.

An attached is a response to comments.

Best regards,

Keio University
School of Integrated Design Engineering
Akihiro Okano
E-Mail: [email protected]
